# Role of Graphene in Constructing Multilayer Plasmonic SERS Substrate with Graphene/AgNPs as Chemical Mechanism—Electromagnetic Mechanism Unit

**DOI:** 10.3390/nano10122371

**Published:** 2020-11-28

**Authors:** Lu Liu, Shuting Hou, Xiaofei Zhao, Chundong Liu, Zhen Li, Chonghui Li, Shicai Xu, Guilin Wang, Jing Yu, Chao Zhang, Baoyuan Man

**Affiliations:** 1Collaborative Innovation Center of Light Manipulations and Applications, School of Physics and Electronics, Institute of Materials and Clean Energy, Shandong Normal University, Jinan 250358, China; 2018020529@stu.sdnu.edu.cn (L.L.); 201809020414@stu.sdnu.edu.cn (S.H.); 2018010048@stu.sdnu.edu.cn (X.Z.); 2019010052@stu.sdnu.edu.cn (C.L.); lizhen19910528@163.com (Z.L.); 2015020674@stu.sdnu.edu.cn (C.L.); 2019020521@stu.sdnu.edu.cn (G.W.); yujing1608@126.com (J.Y.); 2Shandong Key Laboratory of Biophysics, Institute of Biophysics, Dezhou University, Dezhou 253023, China; shicaixu@dzu.edu.cn; 3Institute for Integrative Nanosciences, IFW Dresden, Helmholtzstraße 20, 01069 Dresden, Germany

**Keywords:** graphene, Ag nanoparticles, multi-layer, SERS

## Abstract

Graphene–metal substrates have received widespread attention due to their superior surface-enhanced Raman scattering (SERS) performance. The strong coupling between graphene and metal particles can greatly improve the SERS performance and thus broaden the application fields. The way in which to make full use of the synergistic effect of the hybrid is still a key issue to improve SERS activity and stability. Here, we used graphene as a chemical mechanism (CM) layer and Ag nanoparticles (AgNPs) as an electromagnetic mechanism (EM) layer, forming a CM–EM unit and constructing a multi-layer hybrid structure as a SERS substrate. The improved SERS performance of the multilayer nanostructure was investigated experimentally and in theory. We demonstrated that the Raman enhancement effect increased as the number of CM–EM units increased, remaining nearly unchanged when the CM–EM unit was more than four. The limit of detection was down to 10^−14^ M for rhodamine 6G (R6G) and 10^−12^ M for crystal violet (CV), which confirmed the ultrahigh sensitivity of the multilayer SERS substrate. Furthermore, we investigated the reproducibility and thermal stability of the proposed multilayer SERS substrate. On the basis of these promising results, the development of new materials and novel methods for high performance sensing and biosensing applications will be promoted.

## 1. Introduction

Surface-enhanced Raman scattering (SERS) is a powerful detection and analysis tool that can detect low-concentration molecules with an enhancement factor of 10^14^ or higher [1,2,3,4,5], exhibiting wide application in biomedicine [6], catalytic monitoring [7], environmental analysis [8], food safety [9], and other fields. Although the complete mechanism of SERS is still under debate, two enhancement mechanisms are widely accepted to explain the SERS effect: electromagnetic mechanism (EM) and chemical mechanism (CM) [10,11]. It is generally believed that hot spots are the main contribution for the EM [12,13]. As the main component of the hot spot, Au and Ag nanostructures with strong plasmon resonance coupling and high SERS activity have been widely used and investigated [14,15]. In particular, AgNPs have received more attention due to their superior plasmonic features [16]. CM is generated by molecular resonance excitation or a new charge transfer state generated by metal and adsorbate molecules [17]. 

Since the interesting discovery of graphene-enhanced Raman scattering in 2010, graphene and graphene-based substrates have attracted great attention the theoretical research and application exploration of surface-enhanced Raman scattering (SERS). This makes SERS involving graphene an emerging field in many scientific branches. In particular, benefitting from the outstanding advantages compared with the traditional materials, graphene has proven to be an amazing SERS-active substrate [18,19]. Graphene has a large surface area and excellent molecular adsorption capacity, which can effectively quench the photoluminescence of fluorescent dyes and completely eliminate the fluorescent background [20,21]. By modifying AgNPs on the surface of graphene, one can achieve the coupling of the EM effect of AgNPs and the CM effect of graphene, introducing an excellent SERS performance. Up until now, many different types of the AgNPs and graphene hybrids have been constructed and employed as SERS substrates for sensing inorganic ions, dye molecules, pesticides, and DNA molecules [22,23]. Interestingly, the deposition of closely spaced AgNPs on copper using a graphene island template can achieve an improvement in SERS performance. Irregular shaped AgNPs can provide additional geometric field enhancement like that between two needle-like metal tips [24]. However, the low-dimensional nanostructures that transfer AgNPs to graphene limit the generation of high-density hot spots and poor controllability. Although the sandwich structure constructed by graphene between two layers of AgNPs can enhance the local field, the hot spot density has room for further improvement. On the basis of this, a multilayer structure with high density and high strength hot spots standing on the two-layer metal nanostructure has been developed. The most popular recent research is the vertical structure of multilayer metal nano-arrays, which can greatly increase the density of hot spots due to the longitudinal extension. Li et al. used graphene oxide(GO) as the spacer layer of the annealed AgNP film to prepare a multilayer structure, and the enhancement factor (EF) of the substrate to rhodamine 6G (R6G) reached 7 × 10^8^ [3]. Anderson et al. assembled multilayer AuNPs on colloidal spheres to control the plasmon resonance wavelength, and the detection limit for adenine molecules reached 10 nM [25]. Li et al. prepared a multilayer structure of Au@probe@SiO_2_ with an enhancement factor of 10^7^ [26]. Nevertheless, the shape, size, and distribution of AgNPs obtained by annealing are hard to control, and it remains difficult to realize the uniform nanoscale gaps between AgNPs. The SERS enhancement effect is unstable because of the inhomogeneous distribution of nanogap in the dispersed NPs system. For a nanofilm, that is, a dense layer of close-packed particles, this limitation is overcome through the aid of uniform nano-gap [27]. In general, the density of nanogap is a critical parameter for SERS activity. The interparticle distances determine the coupling strength. On the other hand, the preparation method is difficult, and resulting large-area preparation cannot be realized. Moreover, the synthesis methods of the hybrid nanostructures should be simple enough to potentially practical applications.

In this work, we combined graphene and AgNPs into a CM–EM unit with ex situ self-assembly method and constructed the small unit into a multilayer structure, attempting to improve the SERS performance. The proposed multilayer structures with the CM–EM unit have several advantages: (1) the CM–EM unit can be directly synthesized on any substrate by the self-assembly process, which is simple and can be considered for mass production; (2) the graphene can provide a platform for symmetrical AgNP large-area deposition and ensure a well-proportioned distribution of nanogaps between denser hot spots, which can endow higher EM enhancement. The multilayer structure based on CM–EM unit substrate exhibited excellent detection capabilities for R6G and crystal violet (CV) molecules at low concentration. Moreover, we investigated the excellent reproducibility and thermal stability. Therefore, we expect that this high-performance SERS multilayer substrate can open up a novel method for high performance sensing and biosensing applications.

## 2. Experiment

### 2.1. Materials and Instruments

Acetone (CH_3_COCH_3_, 99.5%), ethylene glycol (C_2_H_6_O_2_, 99.0%), alcohol (C_2_H_6_O, 99.7%), hexane (C_6_H_14_, 97%), and ferric chloride (FeCl_3_) were purchased from a local chemical plant. Polyvinylpyrrolidone (PVP, Mw = 55,000) was purchased from Sigma-Aldrich. Silver nitrate (AgNO_3_) was purchased from Aladdin Co., Ltd. Rhodamine 6G (R6G, AR, 25 g) and crystal violet (CV, AR, 25 g) were purchased from Aladdin industrial corporation (Shanghai, China).

### 2.2. Preparation of the SERS Substrate with Different CM–EM Units

The quartz substrates were initially cleaned with ultrasonic processing in acetone, alcohol, and deionized water for half an hour separately to remove the surface contamination. Ag colloids were synthesized according to the method described by our previous work [28]. The graphene was synthesized on copper foil by chemical vapor deposition (CVD), as reported in detail in our previously study [29]. Next, we dissolved FeCl_3_ (81 g) in 300 mL of deionized water under stirring. Then, the copper foil after the deposition of graphene was placed in the FeCl_3_ solution for 8 h to etch the Cu foil away. After completely removing Cu, we used a clean quartz substrate to gently drag the graphene film out to deionized water to keep it floating and to soak it for 10 min. After the move to clean ionized water and soaking for 10 min, we repeated the process three times to ensure that the residual ferric chloride was removed. The AgNP plasmonic films were prepared by self-assembly at the liquid–liquid interface using the solution of AgNPs with hexane and ethyl alcohol with a volume ratio of 2:1:1. A total of 10 mL Ag colloids and 5 mL of hexane were mixed. Then, 5 mL ethanol was gradually dropped into the above solution. With the increase of the added ethanol, a layer of AgNP film could be clearly seen covering the hexane–Ag colloid interface. After hexane was evaporated, the AgNP film was obtained and removed with a graphene-based quartz substrate. Repeating the step as shown in Figure 1, the multilayer substrate with different CM–EM units can be fabricated.

### 2.3. Characterization

The morphologies of the prepared samples were characterized by scanning electron microscope (SEM, ZEISS Sigma500) with energy dispersive spectrometer (EDS). All the SEM images in the paper were measured at 3 kV voltages. The more detailed morphology and composition were characterized by transmission electron microscope (TEM, JEM-2100F) and high-magnification transmission electron microscopy (HRTEM).

The absorbance spectra of the AgNPs were obtained with a spectrophotometer (PERSEE, TU-1900) using the absorbance mode. The spectrum range was typically from 300 to 800 nm.

SERS spectra were detected by Raman spectrometer (Horiba HR Evolution 800) with a laser wavelength of 532 nm. The laser excitation energy and spot were 0.48 mW and 1 μm, respectively. Throughout the experiment, the diffraction grid was set as 600 gr/mm and the integration time was set as 8 s. The laser light was coupled through an objective lens of 50×.

### 2.4. FDTD Simulations

The electromagnetic field distributions were simulated with finite-difference time domain (FDTD) simulation. In theoretical simulations, the absorption boundary condition is the perfect matching layer (PML). Cross-stack spherical AgNPs that were obtained from SEM with diameters of 57 nm and a 3 nm gap were stacked along the *x-*, *y-*, and *z*-directions, and the 532 nm incident wave polarized along the *x*-direction was set. Considering that graphene does not remain flat when transferred to AgNPs, the graphene in simulation model was set as a curved surface. The numerical data of the refractive index of Ag and graphene were obtained from the reported works [30,31,32].

## 3. Results and Discussion

Graphene can provide an excellent nano-platform for the manufacture of SERS active substrates due to its two-dimensional (2D) planar structure. Figure 2a presents the Raman spectrum of the single layer graphene, where the D, G, and 2D peaks of graphene located near 1350, 1580, and 2700 cm^−1^ can be observed, respectively. The intensity ratio of the G peak to the 2D peak was about 0.5; the full width at half maximum of the 2D peak was less than 35 cm^−1^; and the intensity of the D peak was weak, implying that high-quality and monolayer graphene was successfully transferred to the substrate [33,34]. To investigate the role of graphene for AgNP large-area depositing, we performed a contrast experiment on the substrate with and without the support of graphene in Figure 2b. It can be seen clearly the AgNPs distributed more uniformly on the region with graphene support compared with the region without graphene. Moreover, the AgNPs supported by graphene were closely arranged, forming a large-area stable and uniform film structure. Different from the bare substrate, the AgNP film can be facially transferred to the flexible graphene film substrate, and it tightly adheres to the graphene film surface via van der Waals force [35]. Figure 2c shows the TEM image of the AgNPs with a diameter of about 57 nm. Moreover, the HRTEM inset in Figure 2d was measured to further analyze the details of AgNPs, where the two distinct inter-layer spacings with values of 0.24 nm were agreement with the (111) plane of AgNPs. The absorption peak of AgNPs was approximately 437 nm, as shown in Figure 2d, which matched well with the incident light and was beneficial for generating a stronger local electric field.

SEM was carried out to investigate the surface morphology of the prepared multilayer substrate with different CM–EM units. The image shown in Figure 3a exhibits the self-assembled AgNPs’ firmly ordered arrangement on the surface of the graphene, which is beneficial for the uniform hot spots. The SEM images shown in Figure 3b–f carefully chosen at the boundary of the multilayer substrate clearly present the distribution of AgNPs above and below the graphene. It can be clearly seen that each layer self-assembled AgNP film was compact and uniform, with no visible stacking, and the AgNPs covered with graphene were relatively dark. It is worth mentioning that when the number of units was more than three, the difference in the distribution of AgNPs was almost invisible. The EDS spectrum shown in Figure 3g provides further credible evidence for the successfully fabrication of the multilayer substrate.

To investigate the effect of the CM–EM unit for the SERS performance of the multilayer SERS substrate, we used R6G with the concentration of 10^−6^ M as the probe molecule. Figure 4a shows the SERS signal of R6G collected from the CM–EM unit substrate with different numbers of units. All the typical Raman peaks of the R6G at 613, 774, 1365, 1510, and 1652 cm^−1^ were observed, which was consistent with previous reports [36]. Figure 4c plots the intensity of the SERS signal at 613, 774, and 1365 cm^−1^ for R6G. It can be observed that the Raman signal of R6G firstly increased and then decreased as the number of units increased, reaching a maximum intensity in the four units. As the CM–EM units were successively stacked, the strong plasmnoic coupling between the AgNPs and the coupling between the graphene and the AgNPs were dually excited, and were further synergistically enhanced in the unit. In addition, the introduced graphene also benefitted from the amplification of the SERS signal with the assistance of CM. However, the reason for the intensity of R6G SERS signal remaining almost invariable for more than four units was potentially due to the laser penetration depth, wherein an upper limit for the coupling between different units exists. To better understand this, we built a simulation to explore electric field distributions for multilayer substrate with 1–6 units. As shown in Appendix A, we observed that the intensity of the maximum electric field increased continuously as the number of units increased, and the maximum electric field intensity tended to be stable for more than four units, which indicates that the multilayer structure existed in an upper limit for coupling. However, the Raman signal of R6G reached its maximum value on four units in the experiment, which was ascribed to the inconsistent penetration depth of the incident light in the simulation and experiment. It is worth noting that due to the influence of the curved surface of the graphene, the particles may not be round. On the other hand, the interparticle distances determined the coupling strength. The distance between the lower unit and the higher unit increased as the units were superimposed. Therefore, each of the EM units were plasmonically decoupled between higher and lower units. It should be emphasized that the molecules were adsorbed on the top unit of the multilayer SERS substrate due to the presence of the graphene. However, it should be recognized that the electromagnetic enhancement effect of the AgNPs would not be shielded by the interlayer graphene, and the charge transfer between graphene and probe molecule could produce CM enhancement. In order to prove the universality and reliability of this conclusion, we repeated the experiment with CV and found that the same conclusion can be drawn to prove our point (the relevant results are shown in Figure 4b,d). Interestingly, we also observed the same phenomenon in Figure 4b—the intensity of the SERS signal also increased up to four units stacked. Then, the intensity decreased and finally remained almost unchanged when there were more than four units. The variation of average intensity of CV bands at 914, 1175, and 1619 cm^−1^ relative to the number of the units was also exhibited, as shown in Figure 4d. The transmittance of the multilayer substrate with different units was observed, as shown in Figure 4e. We can observe clearly that the transmittance of multilayer substrate remarkably decreased as the number of units increased, which indicates the attenuation of the incident light from layer to layer and was the reason for the different SERS activity of the substrate with different CM–EM units. On the other hand, the effective penetration depth of incident light is also another reason for the phenomenon of the Raman signal of R6G not continuing to rise when there were more than four units. The laser penetration depth was proportional to the wavelength, and different wavelengths of light penetrated the different depths of substrate. The penetration depth was estimated by the following equation: $d=\frac{\lambda_{1}}{2\pi {\left[\sin^{2}i-{\left(n_{2}/n_{1}\right)}^{2}\right]}^{\frac{1}{2}}}$, where λ_1_ is the wavelength of the light beam in medium 1; *i* is the angle of incidence; and n_1_ and n_2_ are the refractive indices of medium 1 and medium 2, respectively. Because the refractive index of the sample was not only related to the incident wavelength but also to the degree of light absorption of the sample, the refractive index of the sample changed drastically where strong absorption occurred. Therefore, the maximum penetration depth of the 532 nm laser was about 266 nm. Moreover, there was also a maximum collection depth for the Raman detector. Combined with the transmittance of the multilayer structure, the loss of gain for more than four units could result from the lower efficiency of light reaching the lower unit. Hence, the maximum enhancement effect was achieved on the four units. Above all, the multilayer substrate with four unit possessed the optimal SERS performance, which was maintained for the further research throughout the following experiments.

To further investigate the Raman properties of multilayer nanostructure, we directly fabricated multilayer substrate with four CM–EM units. R6G and CV molecules were successively diluted with water solution from concentration of 10^−^^6^ M to 10^−^^14^ M and from 10^−^^5^ M to 10^−^^12^ M, respectively. Then, 2 μL diluted solution was dropped on the surface of the substrate and dried up naturally before SERS detection. The substrate exhibited superior detection capacity for analytes with ultra-low concentration. Raman spectra of R6G and CV molecules collected from the substrate are shown in Figure 5a,b. The characteristic Raman peaks of R6G and CV for the concentration from 10^−14^ M to 10^−6^ M and from 10^−12^ M to 10^−5^ M, respectively, can clearly been observed. The intensities of the SERS signals of the R6G and CV molecules gradually declined with the decrease of the R6G and CV concentration. The quantitative detection can be achieved for R6G and CV. The linear fit curves under the log scale of the vibrations of R6G located at 613, 774, and 1365 cm^−1^ and the vibrations of CV located at 914, 1175, and 1619 cm^−1^ relative to the concentration were exhibited, as shown in Figure 5c,d. The correlation coefficient (*R^2^*) was 0.994 for 613 cm^−1^, 0.995 for 774 cm^−1^, and 0.994 for 1365 cm^−1^. Meanwhile, Figure 5d presents high *R^2^* of 0.989, 0.992, and 0.990 for CV molecules at 914, 1175 and 1619 cm^−1^, respectively, in log scale between the intensity of SERS signal and the concentration. Thus, a good linear relationship between the concentration and the Raman intensity demonstrated the potential for the quantitative detection of chemicals.

The enhancement factor (EF) is an effective method to evaluate the contribution of the proposed multilayer SERS substrate from the enhanced Raman spectra of R6G molecules. The EF is estimated according to the comparison of the limit Raman signal collected from multilayer substrate with four CM–EM units that were obtained from ordinary substrate of probe molecules. The EF was estimated by the following equation [37]: $\mathrm{EF}=(I_{SERS}\times N_{SiO_{2}})/(I_{SiO_{2}}\times N_{SERS})$, where I_SERS_, $I_{{\mathrm{SiO}}_{2}}$, $N_{{\mathrm{SiO}}_{2}}$, and N_SERS_ represent the intensity of the SERS signal, the Raman signal intensity obtained from SiO_2_, the number of analyte molecules within the laser spot on the SiO_2_ substrate, and the number of molecules within the laser spot on the SERS substrate, respectively [38]. To scientifically guarantee the results, we chose the 10^−1^^3^ M R6G solution as the limit concentration for EF calculation. Similarly, the 10^−1^^0^ M CV was also chosen as the limit concentration for the calculation of the enhancement factor. The reason for this choice is because the concentration belongs to the same level of precision (around 6%) in relative standard deviation (RSD), as shown in Appendix A. The Raman signals of R6G and CV with concentration 10^−2^ M obtained from SiO_2_ substrate for reference are shown in Figure 6a. The EF of the multilayer SERS substrate was calculated to be 5.86 × 10^10^ for R6G and 9.62 × 10^8^ for CV, which is better than that of the previously reported SERS substrates with similar multilayer structure [3,39,40,41,42,43,44].

The homogeneity and stability are also significantly essential for SERS substrates. Figure 6b shows the SERS mapping at 613 cm^−1^ peak of 10^−6^ M R6G on multilayer substrate with four CM–EM units in an area of 20 × 20 μm^2^. The step-size of the SERS spectra collection is 2 μm. Clearly, it can be concluded that there was good homogeneity from the relatively smooth and uniform color distribution, with only a small dark region. Reproducibility is a significant parameter for practical application. Figure 6c presents the SERS spectra of the R6G with a concentration of 10^−6^ M collected from 10 different batches of multilayer substrate with four CM–EM units, with each spectrum being the average of the 20 random spots collected from each substrate. The spectra of R6G were greatly well consistent with each other and the intensities for various peaks only fluctuated quite mildly. As demonstrated in Figure 6d, the intensities of the three main R6G characteristic peaks at 613, 774, and 1365 cm^−1^ were collected and the RSDs were 3.52%, 6.22%, and 5.92%, respectively. A similar phenomenon was observed for CV with a concentration of 10^−5^ M in Figure 6e. Moreover, the RSDs of CV at 914, 1175, and 1619 cm^−1^ peaks were 3.26%, 3.01%, and 4.46%, respectively, as shown in Figure 6f. In addition, the RSD on low concentrations of R6G and CV is shown in Appendix A. The RSDs of R6G at 10^−12^ M, 10^−13^ M, and 10^−14^ M were calculated as the values of 5.83%, 6.23%, and 6.87%, respectively, and the RSDs of CV at 10^−10^ M, 10^−11^ M, and 10^−12^ M were 5.83%, 6.23%, and 6.87%, respectively. The RSDs of R6G and CV were less than 10% at low concentrations, which indicates the excellent capability of substrate in SERS detection. These results demonstrate the excellent reproducibility of the multilayer SERS substrate. The expected uniformity and reproducibility could be ascribed to the well-distributed AgNPs and the existence of graphene films in CM–EM units. The well-arranged AgNPs can provide dense electromagnetic hot spot distribution from different units, and graphene can make molecules be well distributed around hot spots, allowing molecules to contact well with the substrate in consequence. Furthermore, the plasmonic coupling between in-plane AgNPs and graphene in each unit can be generated under the motivation of the laser. Additionally, the colloidal self-assembly method was low-cost, simple, and used repeatable techniques that could contribute to excellent reproducibility and make the multilayer SERS substrate with CM–EM unit possess great potential for real applications.

In order to further understand the SERS activity of the multilayer SERS substrate with CM–EM units, we carried out a control experiment to measure Raman spectra on different types of substrate, as depicted in Figure 7a,b. All typical Raman peaks of R6G are shown in Figure 7a, and it is evident that intensities of Raman spectra on the four CM–EM unit substrates were much stronger than others. Figure 7b indicates the intensity of the SERS signal at 613, 774, and 1365 cm^−1^ on the different types of substrate, and the intensities on the four CM–EM unit substrates were about 1.4–3 times stronger than others. As shown in the SEM image in Figure 7c, the arrangement of AgNPs on four EM units was all in a muddle due to the absence of graphene, which can lead to the molecules keeping away from hot spots, whereas the reason for the enhancement being relatively weak was due to the poor binding force between the AgNPs and the detection molecule. Moreover, the molecule was difficult to adsorb on the metal surface, resulting in worse SERS signal stability. As shown in Figure 7d, the distribution of AgNPs on the four EM–CM unit substrates was similar to that on the four CM–EM unit substrates, which proves that there was no particularly large difference in hot spots among AgNPs on different units. The difference for the SERS activity for the four EM–CM units and the four CM–EM units may have been introduced by the top graphene layer. To further identify the perfect SERS behavior of the multilayer substrate with four CM–EM units, we analyzed the electric field properties of these substrates on the basis of the finite difference time domain method (FDTD). The simulation set-up is shown in Figure 7e. The *x-z* views of the electric field enhancement variation (E/E_0_) of the hot spots versus three types of multilayer substrate are shown in Figure 7f. Note that the intensity of the electric field varied with different structures. The intensity of the maximum electric field continuously increased with the addition of graphene. In order to ensure the reliability of the model, we show in Appendix A the electric field of the particles between layers horizontally offset with respect to the next layer. We note that the field enhancement showed almost no change if the particles in the layer were horizontally offset along the *x-* and *y*-directions with respect to the next layer. The simulation result consisted of our experimental analysis that graphene can effectively enhance the electromagnetic coupling effect. It can be observed clearly that the prominent electric fields were all generated at the nanogaps region produced between the AgNPs, and the electric field strength was further enhanced after adding graphene. We also note that the results in the maximum electric field were not strictly consistent with the experimental results. As a matter of fact, the SERS effect is synergic effect of EM enhancement and CM enhancement and it is incorrectly directly related to the maximum electric field.

Moreover, the presence of graphene can greatly improve the stability of the substrate and endow the substrate with excellent thermal stability. The temperature stability experiments of these three types of substrates are shown in the Figure 8a–c. The SERS signal from four EM unit substrates sharply decreased as the temperature increased, as shown in Figure 8a. The signal of R6G at 200 °C on four EM units dropped to less than 80% of the original intensity, while the signal of R6G at 200 °C on the four EM–CM unit substrates in Figure 8b was dropped to below about 35% of the original intensity. It is evident in Figure 8c that the intensities of R6G spectra at 200 °C on the four-unit CM–EM substrate were relatively stable up to 200 °C, with little change in the intensity of peak. The measured spectra signal on four EM–CM unit substrates decreased more than that of four CM–EM unit substrates because of the lack of protection of the top graphene. Compared with Figure 8d and Figure 7c, we found that the substrate with four EM units after heating underwent tremendous changes. The agglomeration phenomenon occurred in the AgNPs, which resulted in a large loss of hot spots. Therefore, the destruction of the hot spots between the units when the agglomeration occurred was the main reason for the substantially reduced Raman signal. As shown in Figure 8e for the case of the four EM–CM unit substrates, the existence of the graphene layer improved the deformation resistance of the multilayer substrate, and the AgNPs will not agglomerate at a large scale. Figure 8f reveals that the introduction of graphene as the top shielding layer can further alleviate the degree of particle aggregation. The measured spectra signal on four EM–CM unit substrates decreased more than that of four CM–EM unit substrates because of the lack of protection of the top graphene. The phenomenon can be explained by the fact that graphene can provide structural support in the longitudinal direction, making the structure tougher and not easy to deform, as shown in Figure 8f. Moreover, graphene can also isolate the contact between AgNPs and air and achieve an anti-oxidation effect, which can broaden the practical application under harsh conditions. Moreover, graphene on the CM–EM unit substrate is a flexible material that can also expand as the AgNPs move after heating, making the CM–EM structure unbroken, which is conducive to maintaining the electric field coupling effect.

## 4. Conclusions

In conclusion, we proposed a multi-layer hybrid SERS structure with graphene as a chemical mechanism (CM) layer and Ag nanoparticles (AgNPs) as an electromagnetic mechanism (EM) layer, forming a CM–EM unit. We investigated the SERS activity of the multilayer with different types of units experimentally and in theory. We demonstrated that the multi-layer structure with a CM–EM unit can serve as a highly sensitive, uniform, and stable SERS substrate. Meanwhile, the reproducibility and thermal stability were also measured from the multilayer SERS substrate. The proposed strategy for self-assembly of multilayer substrate can pave the way towards SERS, and it will possess great potential in the field of application for biosensing.

## Figures and Tables

**Figure 1 nanomaterials-10-02371-f001:**
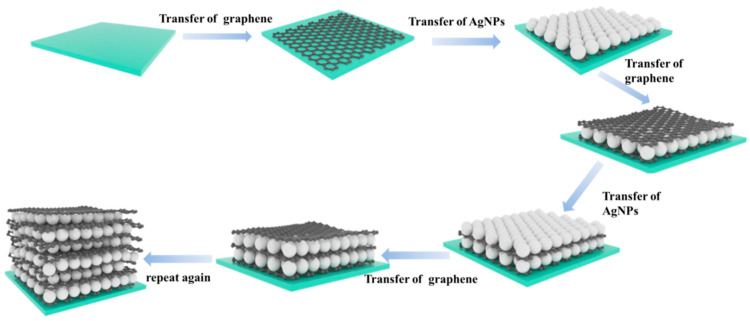
Schematic illustration of the fabrication process of the proposed multilayer substrate.

**Figure 2 nanomaterials-10-02371-f002:**
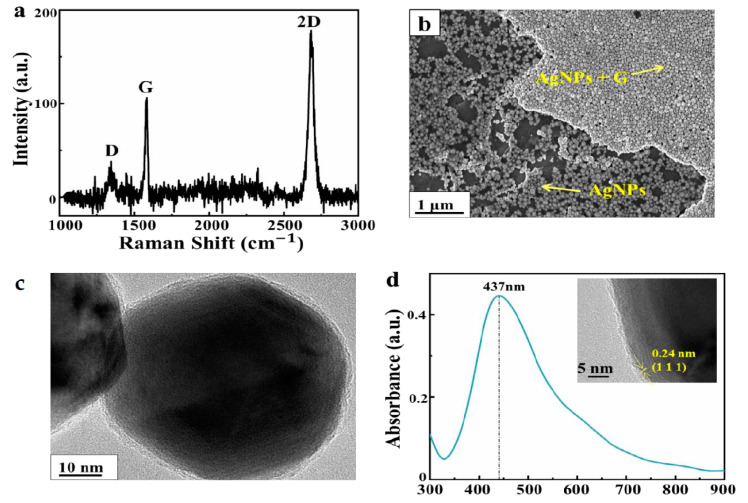
(**a**) Raman spectrum of single layer graphene collected from bare graphene. (**b**) SEM image of the boundary between Ag nanoparticles (AgNPs) and AgNPs supported by graphene. (**c**) TEM image of an AgNP. (**d**) UV−visible absorption spectrum of AgNPs (inset: high−magnification transmission electron microscopy (HRTEM) image of AgNPs).

**Figure 3 nanomaterials-10-02371-f003:**
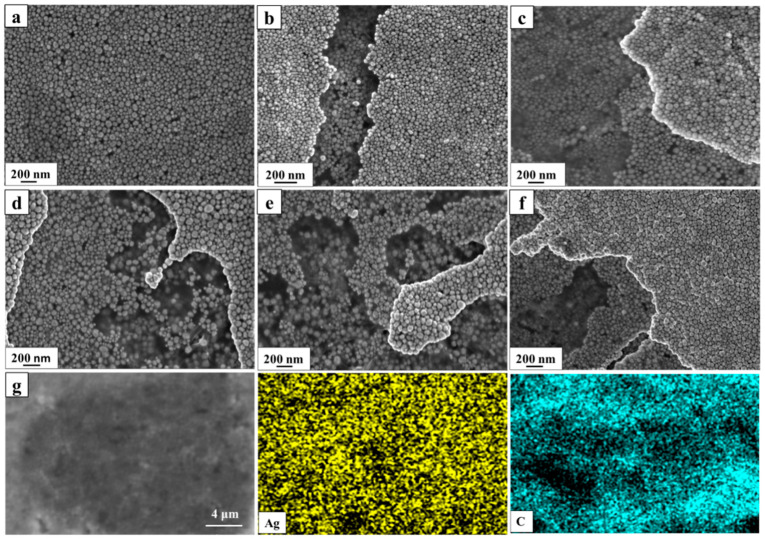
SEM images of the multilayer substrate with (**a**) one chemical mechanism (CM)–electromagnetic mechanism (EM) unit, (**b**) two CM–EM units, (**c**) three CM–EM units, (**d**) four CM–EM units, (**e**) five CM–EM units, and (**f**) six CM–EM units. (**g**) Energy dispersive spectrometer (EDS) elemental maps for the Ag and C on the multilayer substrate.

**Figure 4 nanomaterials-10-02371-f004:**
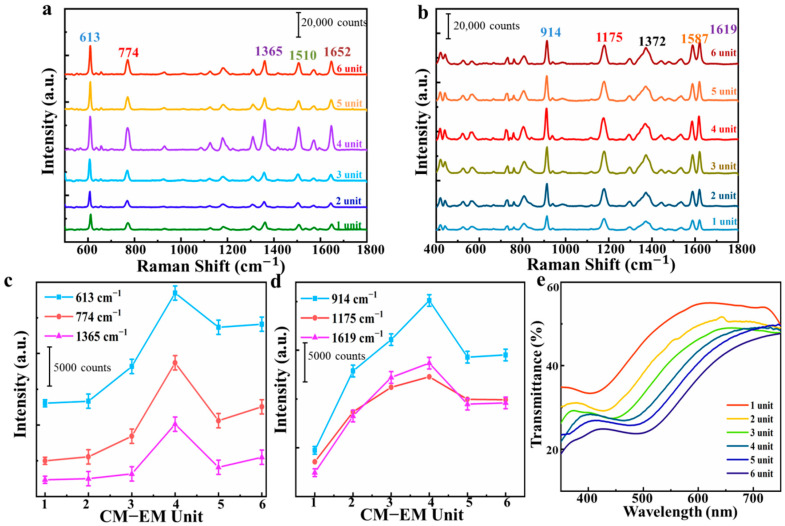
Raman spectra of (**a**) rhodamine 6G (R6G) with concentration of 10^−6^ M and (**b**) crystal violet (CV) with concentration of 10^−5^ M obtained from the multilayer substrate with various CM−EM units. Raman signal intensity of (**c**) R6G with concentration of 10^−6^ M at 613, 774, and 1365 cm^−1^ peak, and (**d**) CV with concentration of 10^−5^ M at 914, 1175, and 1619 cm^−1^ peak. (**e**) The transmittance of the substrate with different units.

**Figure 5 nanomaterials-10-02371-f005:**
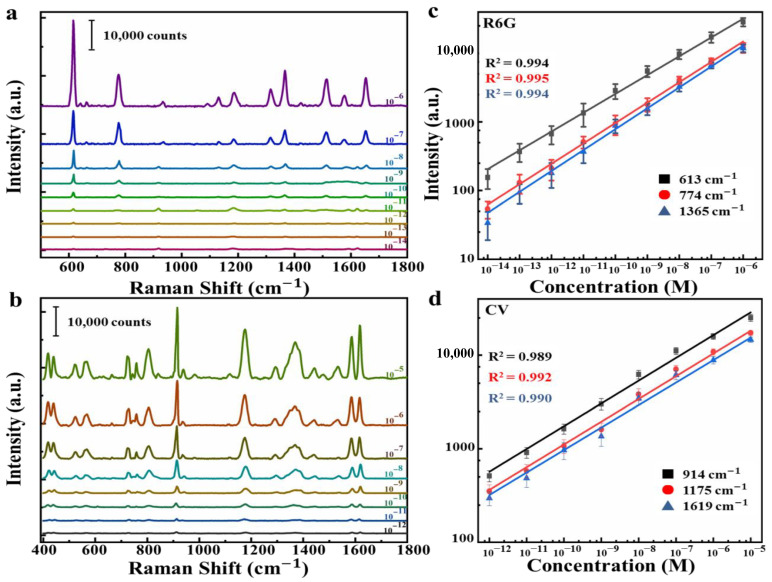
Raman spectra of (**a**) R6G and (**b**) CV on multilayer substrate with four CM−EM units from 10^−14^ M to 10^−6^ M and from 10^−12^ M to 10^−5^ M, respectively. The Raman signal intensity of (**c**) R6G at 613, 774, and 1365 cm^−1^ peak, and (**d**) CV at 914, 1175, and 1619 cm^−1^ peak as a function of molecular concentration in the log–log scale.

**Figure 6 nanomaterials-10-02371-f006:**
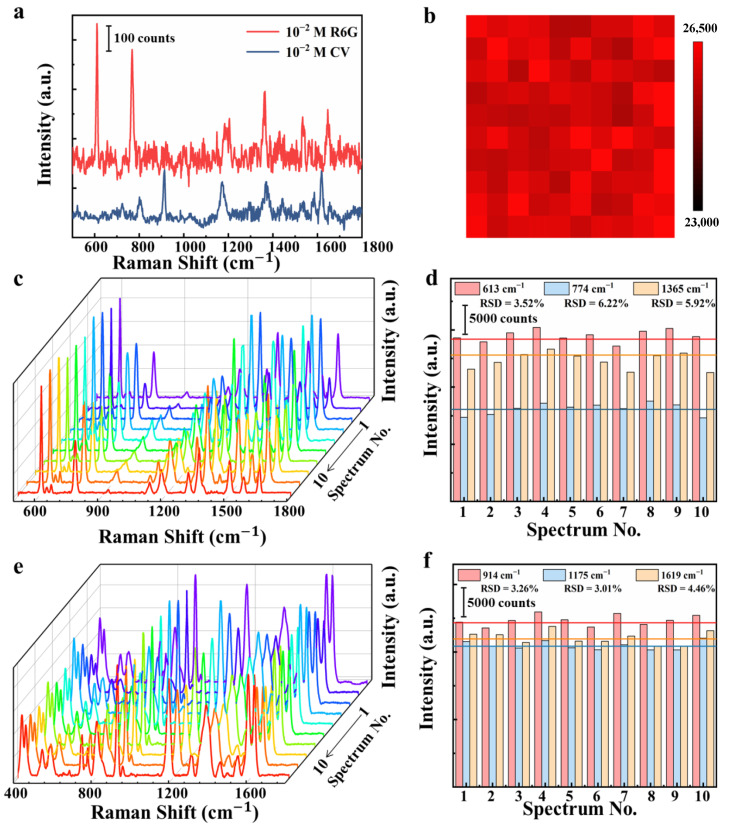
(**a**) Raman spectra of R6G and CV with concentration of 10^−2^ M collected from the SiO_2_ substrate. (**b**) Raman mappings at the 613 cm^−1^ peak in 20 × 20 μm^2^ area collected from multilayer substrate with four CM–EM units. Raman spectra of (**c**) R6G with a concentration of 10^−6^ M and (**e**) CV with a concentration of 10^−5^ M from 10 different batches of multilayer substrate with four CM–EM units. Intensity distribution of the peak at (**d**) 613, 774, and 1365 cm^−1^ for the R6G with a concentration of 10^−6^ M and (**f**) 914, 1175 and 1619 cm^−1^ for the CV with a concentration of 10^−5^ M from 10 different batches of multilayer substrate with four CM–EM units.

**Figure 7 nanomaterials-10-02371-f007:**
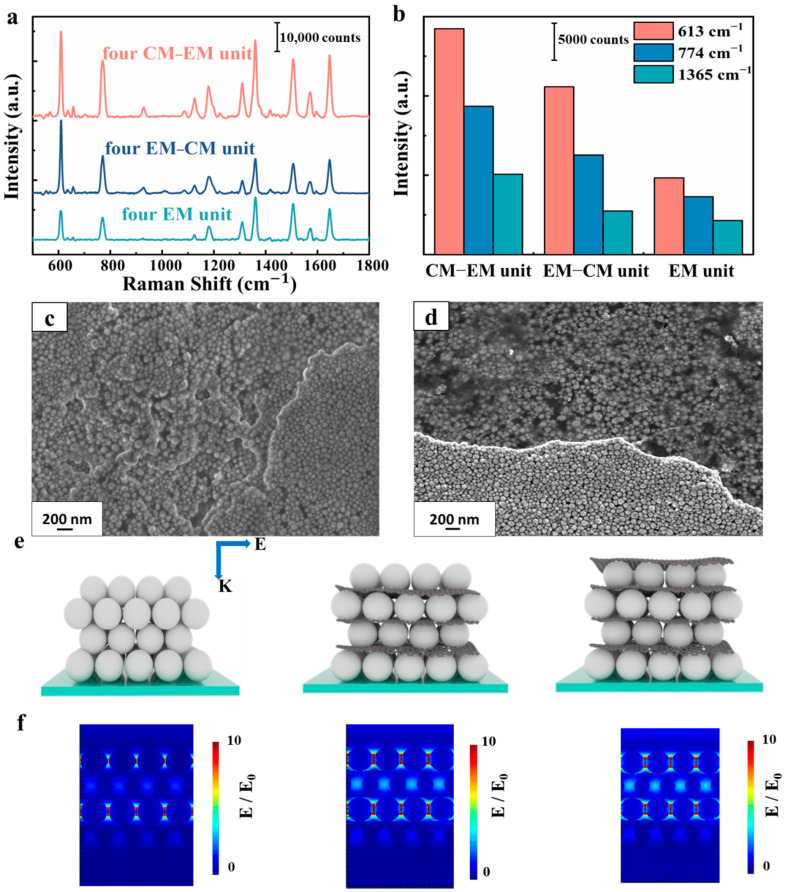
(**a**) Raman spectra of R6G with concentration of 10^−6^ M obtained from four EM unit substrates, four EM–CM unit substrates, and four CM–EM unit substrates. (**b**) Raman signal intensity of R6G at 613, 774, and 1365 cm^−1^ peaks. SEM images of (**c**) four EM unit substrates and (**d**) four EM–CM unit substrates. (**e**) Simulation set-up of four EM unit substrates, four EM–CM unit substrates, and four CM–EM unit substrates. (**f**) The *x-z* views of electric field distribution of different SERS substrates at 532 nm wavelength.

**Figure 8 nanomaterials-10-02371-f008:**
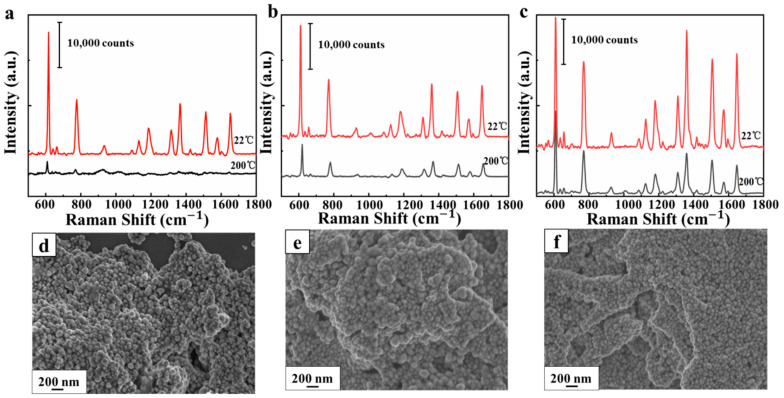
Raman spectra of R6G with concentration of 10^−6^ M on (**a**) four EM unit substrates, (**b**) four EM–CM unit substrates, and (**c**) four CM–EM unit substrates at room temperature and at 200 °C. SEM images of (**d**) four EM unit substrates, (**e**) four EM–CM unit substrates, and (**f**) four CM–EM unit substrates after heating at 200 °C.

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
