# Peer review of "Role of Graphene in Constructing Multilayer Plasmonic SERS Substrate with Graphene/AgNPs as Chemical Mechanism—Electromagnetic Mechanism Unit"

_nanomaterials, 2020, doi:10.3390/nano10122371_

Round 1

Reviewer 1 Report

Authors tried to enhance the sensitivity of self-assembled AgNP SERS sensors by combining electromagnetic enhancement of AgNPs with chemical enhancement of graphene and by stacking multiple layers of AgNPs/Gr. I hope the following comments can help authors revise the manuscript before it is re-considered for publication.

The general concerns are as follows:

  1. If authors can justify the tedious transfer process for graphene and prove it economic, this might be useful.
  2. When graphene is transferred on AgNPs, graphene does not remain flat. The simulation model needs to include errors caused by the nearly conformal coverage AgNPs by graphene.
  3. The reproducibility of stacking multiple layers of AgNPs/Gr determines whether the reported result of achieving the maximum sensitivity of the four-layer sensor is correct or it is just an outcome of one single stacking experiment.
  4. There are recent publication(s) reporting one-layer Ag SERS with much higher sensitivity (lower low detection limit, e.g. 10-16 M R6G, Tzeng et al. in IEEE TNANO 2019 and 10-12 M adenine than what multi-layer SERS sensors can achieve. Authors should cite compare and comment on those publications.  

The remaining details are provided to help with authors revisions:  

  1. “The strong couple between the hybrid of graphene and metal particles can greatly improve the SERS performance and thus broaden the application fields.”
  2. For which kinds of molecules can the graphene and metal particles improve the SERS performance? For all?
  3. Why is thermal stability at 200C important?
  4. How to and how long does it take to transferring graphene?
  5. Fig 5d CV counts at 10-12 M is higher than the R6G count at 10-14 M. Why is the sensitivity for CV (10-12M) lower than R6G (10-14M)?
  6. How is the reproducibility near the low detection limit? Fig. 6 displays CV at 10-5 M and R6G at 10-6 M. Please show similar data for CV at 10-10 – 10-12 M and R6G at 10-12 to 10-14 M.
  7. Please clearly point out where in your results is the evidence that graphene does play chemical enhancement roles?
  8. What is the gap spacing between self-assembled neighbouring AgNPs?

Reviewer 2 Report

The manuscript reports an interesting study on the characterization of a sandwich substrate (alternate graphene and AgNPs) for SERS applications.

The resesearch is carried out competently. The results are interesting and the manuscript should be accepted for pubblication with minor revisions.

The major problem is the almost complete lack of experimental details on the sample preparation. Other researchers must be able to repeat the experiment in order to accept the paper for publication. The Authors must provide complete data (or adequate references) on the methods used for the detachment of the graphen from copper (i.e. the amount and concentration of FeCl3 solution used to remove the copper substrate, expressed of ml of solution at a well defined concentration per gram of copper substrate). Also, details on the graphene  transfer to the quartz plate must be provided (and for the transfer of the AgNPs plasmonic film, too). By the way, Figure 1  seems to be cut to the right side nd some details are missing.  

Finally, absolutely no deteal is given on the method for the sample preparation (i.e. the transfer of the dye solution (which solvent was used?) to the SERS substrate). The Authors should explain if they rinse the SERS plate in a solution of the dye (contact time) or they just drop a small volume (how many microliters?) on the surface of the plate.

Some minor english/typo errors are present:

I'm not an a native english speaking person but I wonder it it is correct to use the word "couple" or "coupling" (lines 13, 48 and following)

line 15 hybird -> hybrid

line 42 remove "in"

line 82 Alorich -> Aldrich

line 210 "To guarantee the scientific nature of the results". Please explain better what is the possible problem that is behind this sentence.

Reviewer 3 Report

The authors present multi-layered hybrid structures of graphene and Ag nanoparticles for SERS applications. In this study, two dyes have been utilized as model analytes (CV and R6G) to benchmark the efficiency of SERS generation. A central question relates to the differences in SERS performance with changes in the number of layers. Apparently, a layer count of 4 gave best results. However, the accompanying explanation is not very convincing. It is to be expected that the efficiency of multilayer systems is directly dependent on the transmission of light though the stack. I recommend the authors to revise their explanation of this phenomenon and the scientific discussion on the excitation of hot spots in stacked layers. Overall, the manuscript’s presentation quality is appropriate and its topic fits well with the journal scope of nanomaterials. I recommend acceptance after major revisions.

Further comments and questions:

1) The transfer of AgNPs onto the graphene layer needs to be described in more detail. Does the process produce particle monolayers? Are the particles close-packed?

2) What are the optical properties (transmission UV/vis spectroscopy) of the hybrid structure? This data needs to be presented for the 1, 2, 3, 4, 5, and 6 stacked layer systems. What is the attenuation of light from layer to layer? This will allow to clarify how much light reaches the deeper layers.

3) Each of the layers of AgNPs is plasmonically decoupled from higher/lower levels. This needs to be discussed in more detail to avoid misunderstandings.

4) The additional gain of multilayers could be expected to simply scale with the number of layers. Is this the case in the FDTD simulations?

5) The authors should include in their discussions the amount of light that reaches each layer. As an alternative explanation, this might clarify why the efficiency of enhancement increases up to 4 layers and decreases for 5 and 6 layers. I am convinced that the transmittance of light though each of the CM-EM units needs to be considered here. As such, the loss of gain for more than 4 layers could result from the lower efficiency of light reaching lower layer (without being absorbed by upper layers).

6) The title is not appropriate because many readers might not be familiar with the term CM-EM unit.

7) In Fig. 6b, the scaling of the color scale is not clear.

8) Concerning the simulations (Fig. 7): How does the field enhancement change if the particles in the layer are offset (in respect to the next layer)?

9) In Fig. 4, the color code of the text labels in a and b are confusing. Shouldn’t these be matched to the colors in c and d?

10) Subfigure 1 of 7e looks vertically stretched.

11) Subfigures in 7f look horizontally stretched. I suggest the same color scaling to allow for a better comparison.

12) Concerning the FDTD data, how was the presence of graphene simulated?

13) How was the heating performed? It can be expected that heating in inert atmosphere or at air gives different results.

14) Further proof reading might be advisable to remove typos and bad wording: p.4 van der Waals; p.9 reviously reported; p.8 “to guarantee the scientific nature of the results”; p.2 “exciting SERS performance”; etc.

15) The authors should discuss the difference between randomly close-packed particle layers and ordered particle assemblies for SERS applications (compare DOI:5772/intechopen.79055).

Reviewer 4 Report

The article is devoted to the development and research of multi-layer hybrid SERS structure with the graphene layer with a chemical mechanism and Ag nanoparticles layer with an electromagnetic mechanism of Raman scattering enhancement. It was shown that such a structure demonstrates a high enhancement factor, which is uniformly distributed over substrate surface and thermally stable in case of four layer CM-EM morphology. Such structure possess great  potential in the field of high performance sensing and  biosensing applications.

My comments on the text of the article are as follows:

  1. The explanation proposed by the authors for the dependence of the enhancement factor (EF) on the number (N) of CM-EM layers explains only the saturation of the EF; at the same time, we observe a pronounced unexplained maximum at N = 4.
  2. It is not entirely clear how rigidity, resistance to deformation and flexibility of the proposed structure provides an increase in the thermal stability of its enhancement factor?

I would like to receive answers to these comments in the new edition of the article

Round 2

Reviewer 1 Report

  1. Authors have responded to all comments.
  2. “the deposition of closely spaced AgNPs on the graphene island template” should be “the deposition of closely spaced AgNPs on copper using a graphene island template”.
  3. Please add explaination and justification in the manuscript on how molecules can penetrate through layers of high-quality graphene to become affected by the lower layers of a stacked AnNP/Gr 3-D structure. If penetration is not needed, please state that molecules adsorb on the top layer AgNPs/Gr of the stacked SERS sensors.

Reviewer 3 Report

The authors have revised the manuscript according to the comments. Few points have not been resolved. Nevertheless, I recommend publication after minor revisions as indicated below.

1) Fig. S1 needs revisions. The models shown in Fig S1a need to explained in the caption. Also, Fig S1b is missing labeling on the y axis. Therefore, it is not clear how much the field is enhanced. I recommend plotting field strength Eˆ2/E0ˆ2 either on linear or logarithmic scale, whichever seems more suitable.

2) The discussion in Response 5 should be integrated into the main text.

3) The discussion in Response 8 should be integrated into the main text and the Supporting Information.

4) The correct spelling is „Van der Waals“.

5) In Response 14, the authors indicate a concentration of “10-12M R6G“ was used. However, in the manuscript it is written that „10-13 M R6G“ on page 9. This raises questions about the validity of these numbers.

Reviewer 4 Report

The new version of the article looks much better than the previous one and it can be published in its present form

Author Response

Thank you for thoroughly reviewing our manuscript and making the careful and thoughtful comments.